# Molecular Antioxidant Properties and In Vitro Cell Toxicity of the *p*-Aminobenzoic Acid (PABA) Functionalized Peptide Dendrimers [note 1]

**DOI:** 10.3390/biom9030089

**Published:** 2019-03-05

**Authors:** Marta Sowinska, Maja Morawiak, Marta Bochyńska-Czyż, Andrzej W. Lipkowski, Elżbieta Ziemińska, Barbara Zabłocka, Zofia Urbanczyk-Lipkowska

**Affiliations:** 1Institute of Organic Chemistry Polish Academy of Sciences, 01-224 Warsaw, Poland; marta.sowinska@selvita.com (M.S.); maja.morawiak@icho.edu.pl (M.M.); 2Mossakowski Medical Research Centre Polish Academy of Sciences, 02-106 Warsaw, Poland; marta.bochynska@interia.eu (M.B.-C.); andrzej.lipkowski@imdik.pan.pl (A.W.L.); elziem@imdik.pan.pl (E.Z.); bzablocka@imdik.pan.pl (B.Z.)

**Keywords:** dendrimers, PABA, antioxidant, ROS, DPPH, ABTS, melanoma, cerebellar granule cells, Glu

## Abstract

**Background:** Exposure to ozone level and ultraviolet (UV) radiation is one of the major concerns in the context of public health. Numerous studies confirmed that abundant free radicals initiate undesired processes, e.g. carcinogenesis, cells degeneration, etc. Therefore, the design of redox-active molecules with novel structures, containing radical quenchers molecules with novel structures, and understanding their chemistry and biology, might be one of the prospective solutions. **Methods:** We designed a group of peptide dendrimers carrying multiple copies of p-aminobenzoic acid (PABA) and evaluated their molecular antioxidant properties in 1,1′-diphenyl-2-picrylhydrazyl (DPPH) and 2,2′-azino-bis(3-ethylbenzothiazoline-6-sulphonic acid) (ABTS) tests. Cytotoxicity against human melanoma and fibroblast cells as well as against primary cerebral granule cells (CGC) alone and challenged by neurotoxic sodium glutamate and production of reactive oxygen species (ROS) in presence of dendrimers were measured. **Results:** PABA-terminated dendrimers express enhanced radical and radical cation scavenging properties in relation to PABA alone. In cellular tests, the dendrimers at 100 μM fully suppress and between 20–100 μM reduce proliferation of the human melanoma cell line. In concentration 20 μM dendrimers generate small amount of the reactive oxygen species (<25%) but even in their presence human fibroblast and mouse cerebellar granule cells remain intact Moreover, dendrimers at 0.2–20 µM concentration (except one) increased the percentage of viable fibroblasts and CGC cells treated with 100 μM glutamate. **Conclusions**: Designed PABA-functionalized peptide dendrimers might be a potential source of new antioxidants with cationic and neutral radicals scavenging potency and/or new compounds with marked selectivity against human melanoma cell or glutamate-stressed CGC neurons. The scavenging level of dendrimers depends strongly on the chemical structure of dendrimer and the presence of other groups that may be prompted into radical form. The present studies found different biological properties for dendrimers constructed from the same chemical fragments but the differing structure of the dendrimer tree provides once again evidence that the structure of dendrimer can have a significant impact on drug–target interactions.

## 1. Introduction

Since the first report by Tomalia et al., on synthesis of polyamidoamine dendrimers (PAMAM), these branched polyfunctional polymers with well-defined structure have received a lot of scientific attention due to the prospective high potential in a wide range of medical applications [1,2]. High demand for innovative drugs with the high therapeutic index and negligible side effects directed pharmaceutical research into formulation of the known bioactive molecules with safe, site-specific carrier systems. A wealth of small organic biomolecules and their synthetic counterparts have been served for years as indispensable pharmaceutical tools. Recently, dendrimeric carriers with their nanodimensions and apparent proteins mimicry provide a novel and unique way in which these molecules are presented to the cell surface or cell organelles. Therefore, since the beginning of their history, the polyvalent dendrimers were regarded as well-suited for novel biological applications such as regenerative medicine [3], diagnostics [4], the design of drugs targeting the brain [5,6], and gene delivery [7]. 

A challenging medical problem is diminishing the negative effects of an increased exposition of human population to high ozone levels and ultra-violet radiation. Recent studies confirmed that the presence of free radicals initiates undesired processes, e.g. carcinogenesis or degeneration of cells constituting nervous system [8,9]. In this respect, the response of biological molecules toward oxidizing conditions and knowledge of natural compounds repairing abrogated metabolic pathways is of continuing interest. The potential toxicity of the redox-active nanomaterials has been the subject of much concern in a two-fold context: by undesired disturbance of the metabolic pathways [10,11,12] and on the other side, for the possibility of application of these processes in therapy of various diseases (e.g. as reactive oxygen species (ROS)-inducing drugs) [13,14,15]. 

*p*-Aminobenzoic acid (PABA), belonging to the vitamin B group (also referred to as vitamin Bx or B_10_) is one of the pharmaceutically relevant small organic molecules that has various biological applications [16]. In medicine and cosmetics, it is mostly used as a prophylactic against UV-exposure related skin disorders. PABA is precursor in the synthesis of folic acid, another important vitamin B group component. In mammals where folic acid is not biosynthesized, beyond the external supplementation it is in part supplied by the symbiotic bacteria that produce PABA [17]. Moreover, folic acid receptors that are overexpressed on membranes of rapidly proliferating tumor cells are frequently used for the selective internalization of anticancer drugs (a so called “Trojan horse” strategy) [18]. In plants, PABA is known to trigger a systemic acquired resistance (SAR) against bacterial and viral pathogens [19]. Due to its photosensitivity, PABA has also been proposed as a new fluorescent probe, for the selective measurement of peroxyl radical scavenging (PRS) activity in biological samples [20]. 

The aim of the present study was to design and synthesize macromolecular conjugates containing PABA residues presented to the selected biological systems by peptide dendrimeric carriers. The resulting group of cationic, PABA-functionalized peptide dendrimers was tested for their molecular antioxidant activity in relation to chemical structure and in vitro cell toxicity during interactions with healthy and malignant human skin cells (human melanoma and fibroblast cells) as well as their impact on the rat cerebellar granule cells (CGC) as a model of neuron cells, with respect to cell viability and ROS production. 

In summary, the PABA-terminated dendrimers in concentrations above 20 μM express enhanced radical and radical cations scavenging properties (2,2′-azino-*bis*(3-ethylbenzothiazoline-6-sulphonic acid) (ABTS) and 1,1′-diphenyl-2-picrylhydrazyl (DPPH) tests) in comparison to the PABA alone. The scavenging level depends strongly on the dendrimer chemical structure and the presence of other groups that may be prompted into radical form. In cellular tests, the dendrimers at 100 μM fully suppress and between 20–100 μM reduce the proliferative potency of the human melanoma cell line. In concentration, 20 μM dendrimers generate a small amount of the reactive oxygen species (<25%) but even in their presence human fibroblast and mouse cerebellar granule cells remain intact. Moreover, some dendrimers at this concentration expressed an increased cell viability of fibroblast and CGC cells. The peptide dendrimers containing redox-active PABA residues share some of the functional characteristics of natural antioxidants which in in vitro tests exhibit significant antioxidant and radical scavenging properties but reduce viability of cancer cells probably not via ROS-induced apoptosis. 

The different biological properties for dendrimers constructed from the same chemical fragments but differing in the structure of the dendrimer tree, found in the present studies, provide once again evidence that the structure of the dendrimer can have a significant impact on drug–target interactions.

## 2. Materials and Methods 

### 2.1. General Procedures

All solvents and reagents were of analytical grade and were used without further purification. All solvents were obtained from Sigma-Aldrich (Steinheim, Germany). Low resolution mass spectra (LRMS) were recorded with a Mariner electrospray (ESI) time-of-flight mass spectrometer (PerSeptive Biosystems, Foster City, CA, USA) for the samples prepared in MeOH. The ^1^H-nuclear magnetic resonance (NMR) and ^13^C-NMR spectra were recorded using a Bruker Avance spectrometer (Karlsruhe, Germany) at 500/125 or 400/100 MHz, respectively, using deuterated solvents and trimethylsilyl (TMS) as an internal standard. Chemical shifts are reported as δ values in parts per million, and coupling constants are given in hertz. The optical rotations (**[α]_D_^25^**) were measured at 25 °C with a JASCO J-1020 digital polarimeter (Ishikawa-machi, Hachioji, Tokyo, Japan). Melting points were recorded on a Köfler hot-stage apparatus (Wagner & Munz, München, Germany) and are uncorrected. Thin layer chromatography (TLC) was performed on aluminum sheets with silica gel 60 F254 from Merck (Darmstadt, Germany). Column chromatography (CC) was carried out using silica gel (230–400 mesh) from Merck (Darmstadt, Germany) or Sephadex LH20 (Biosciences, Upsala, Sveden). The TLC spots were visualized by treatment with 1 % alcoholic solution of ninhydrin and heating. 

### 2.2. Synthesis and Characterization of p-Aminobenzoic (PABA)-Terminated Dendrimers

The general strategy for the synthesis of a series of amphiphilic dendrimers containing on the surface *p*-aminobenzoic acid residues involves synthesis of the *tert*-butyloxycarbonyl (Boc)-protected new hydrophobic branching units of the AB_1_B_2_ type, where B_1_ and B_2_ denote alkylamino chains of equal or different length (Scheme 1), followed by coupling them with the *N*^α^ or *N*^ε^ atoms of lysine functionalized at C-terminus with benzylethylamine, tryptamine, or dodecylamine moiety (compounds 8–11, Scheme 2) was described in detail recently [21]. After deprotection of amino groups such core molecules are then reacted with suitably protected lysine (Scheme 3). The synthetic pathway for the preparation of branching monomer with variable arm length of AB_2_ or AB_1_B_2_ structure from 3,5-dihydroxybenzoic acid is shown in Scheme 1. The monomer of AB_2_ type was synthesized via modification of a procedure published by Liskamp et al., yielding di-substituted compound 2 with equal arm length, and mono-substituted product 3 that was later used for synthesis of monomer 5 with non-equal arm length [22]. 

Synthesis of dendrimers with terminal p-aminobenzoic acid residues was performed by coupling of dendrimers 16–19 with 4-(Boc-amino)benzoic acid using EDC/HOBt in DMF and 3.6 equiv. of Et_3_N per one group (Scheme 4). Purification by size exclusion chromatography on Sephadex LH-20 with MeOH as eluent, followed by silica gel column chromatography with CHCl_3_/MeOH (8:1 or 50:1→10:1 gradient system) provided dendrimers 20–23 with 39.1%–60% yield, as white to pale yellow solids without sharp melting point (Table 1). Subsequent N-*Boc deprotection* (HCl in EtOAc, 91.3%–97.7% yield) of dendrimers 20–23 dissolved in minimal volume of MeOH, yielded dendrimers 24–27 as hygroscopic octahydrochlorides (Scheme 4 and Table 1).

Structure of dendrimers 20–27 was fully confirmed by their mass spectrometry (ESI MS) and nuclear magnetic resonance (NMR) spectra. In mass spectra of dendrimers 20–27 main signals represent doubly ionized ions of the [M + 2Na]^2+^ type. The pseudomolecular ions of the [M + Na]^+^ type are also visible but are less intense. On the other hand, for deprotected dendrimers 24–27 the signals of [M + 2H]^2+^ and [M + 3H]^3+^ structure are the most intense. 

Interpretation of ^1^H- and ^13^C-NMR spectra of dendrimers 20–27 was done on the basis of distortionless enhancement by polarization transfer (DEPT) and two-dimensional correlation spectroscopy (COSY) and heteronuclear single quantum coherence (HSQC) correlation experiments. In particular signals of aromatic carbon and hydrogen atoms of 4-(Boc-amino)benzoic acid in dendrimers 20–23, or 4-aminobenzoic acid residues in dendrimers 24–27, are clear and well-separated from the remaining signals and can be used for the structural diagnostics. Complete disappearance in ^13^C NMR spectra signals at 28.0–29.0 ppm [C(*C*H_3_)_3_] and 77.9–81.2 ppm [*C*(CH_3_)_3_] and 1.35–1.50 [C(CH_3_)_3_] signals in ^1^H-NMR spectra indicated complete Boc-deprotection of the amine groups in dendrimers 24–27.

### 2.3. Spectrocscopic Data of New Dendrimers (16–27)

#### 2.3.1. Dendrimer 16 (Obtained from 12)

Dendrimer 16 was obtained as creamy powder from 0.63 g (0.28 mmol) of dendrimer 12 dissolved in 10 mL MeOH after 3h hydrogenolysis: 93.8% (0.45 g).

C_86_H_143_O_19_N_15_, M = 1691.15 g/mol (monoisotopic mass 1690.1). LRMS (ESI, MeOH): 1691.2 [M + H^+^]^+^, 846.1 [M + 2H^+^]^2+^, 564.4 [M + 3H^+^]^3+^, 423.6 [M + 4H^+^]^4+^ - *main signal*.

^1^H-NMR (500 MHz, MeOD): δ = 1.30–1.52 [br m, 20H, 5×γCH_2_
*L-Lys and core*, 3×δCH_2_
*L-Lys*, 2×O-(CH_2_)_2_-C***H_2_***-(CH_2_)_2_-NH] overlapped with 1.40, 1.42 [2s, 36H, 4×C(CH_3_)_3_], 1.54–1.66 [br m, 12H, 2×δCH_2_
*L-Lys and core*, 2×βCH_2_
*L-Lys*, 2×O-(CH_2_)_3_-C***H_2_***-CH_2_-NH], 1.67–1.87 [br m, 10H, 3×βCH_2_
*L-Lys and core*, 2×O-CH_2_-C***H_2_***-(CH_2_)_3_-NH], 2.58 (2t, *J* = 6.9 Hz, 6H, 3×εCH_2_
*L-Lys*), 2.78 (t, *J* = 7.2 Hz, 2H, CH_2_-Ar *PEA*), 3.22 [br m, 6H, εCH_2_
*L-Lys*, 2×O-(CH_2_)_4_-C***H_2_***-NH], 3.36 (m, 3H, C***H_2_***-NH *PEA*, εCH_2_
*core*), 3.51 (m, 3H, O-CH_2_-C***H_2_***-NH, C***H_2_***-NH *PEA*), 3.62 (m, 2H, O-CH_2_-C***H_2_***-NH), 3.85-4.12 (br m, 12H, 4×O-CH_2_, 4×αCH *L-Lys*), 4.47 (m, 1H, αCH *core*), 6.62 (m, 2H, C^4^-H *Ph*), 6.94, 7.01 (2m, 4H, C^2,6^-H *Ph*), 7.12–7.25 (br m, 5H, C^2,3,4,5,6^-H *PEA*).

^13^C-NMR (500 MHz, MeOD): δ = 24.2, 24.3 (γC), 24.4 [2×O-(CH_2_)_2_-***C***H_2_-(CH_2_)_2_-NH], 24.5 (γC), 28.8 [C(***C***H_3_)_3_], 29.9, 30.1 [δC, 2×O-CH_2_-***C***H_2_-CH_2_-***C***H_2_-CH_2_-NH], 32.8, 33.3, 33.4 (βC), 36.5 (***C***H**_2_**-Ar *PEA*), 39.9 (2×O-CH_2_-***C***H_2_-NH), 40.2 [2×O-(CH_2_)_4_-***C***H_2_-NH], 40.6 (εC *core*), 42.0 (CH_2_-NH *PEA*), 42.2, 42.3 (4×εC *L-Lys*), 55.5 (αC *core*), 56.2 (4×αC *L-Lys*), 67.7 (2×O-***C***H_2_-CH_2_-NH), 69.1, 69.2 [2×O-***C***H_2_-(CH_2_)_4_-NH], 80.5 [***C***(CH_3_)_3_], 105.7, 106.0 (C^4^
*Ph*), 106.6, 107.0, 107.1, 107.4 (C^2,6^
*Ph*), 127.4, 129.5, 129.9 (C^2,3,4,5,6^
*PEA*), 137.2, 137.8 (C^1^
*Ph*), 140.4 (C^1^
*PEA*), 157.8 [C=O (Boc)], 161.2, 161.3, 161.4, 161.7 (C^3,5^
*Ph*), 169.6, 169.7 (CONH *Ph*), 174.4, 175.2, 175.5 (CONH).

**[α]_D_^25^** = −17.6 (c 0.75, MeOH). 

**M.p.:** 115–123 °C.

#### 2.3.2. Dendrimer 17 (Obtained from 13)

Dendrimer 17 was obtained as yellow powder from 1.13 g (0.52 mmol) of dendrimer 13 dissolved in 20 mL of MeOH after 10h hydrogenolysis: 1.13 g (0.52 mmol), yield 85.3% (0.725 g).

C_82_H_132_O_19_N_16_, M = 1646.02 g/mol (monoisotopic mass 1645.0). LRMS (ESI, MeOH): 823.5 [M + 2H^+^]^2+^, 549.4 [M + 3H^+^]^3+^ - *main signal*, 516.2 [M - Boc + 3H^+^]^3+^, 412.5 [M + 4H^+^]^4+^. 

^1^H-NMR (500 MHz, MeOD): δ = 1.30–1.47 (br m, 18H, 5×γCH_2_
*L-Lys and core*, 4×δCH_2_
*L-Lys*) overlapped with 1.40 [s, 36H, 4×C(CH_3_)_3_], 1.52–1.75 (br m, 10H, δCH_2_
*core*, 4×βCH_2_
*L-Lys*), 1.77–1.90 (br m, 2H, βCH_2_
*core*), 2.56 (m, 8H, 4×εCH_2_
*L-Lys*), 2.93 (t, *J* = 7.2 Hz, 2H, CH_2_-Ar *TA*), 3.3 (m, 2H, εCH_2_
*core*), 3.40-3.65 (br m, 10H, 4×O-CH_2_-C***H_2_***-NH, C***H_2_***-NH *TA*), 4.02 (br m, 12H, 4×O-C***H_2_***-CH_2_-NH, 4×αCH *L-Lys*), 4.50 (m, 1H, αCH *core*), 6.64 (m, 2H, C^4^-H *Ph*), 6.92-7.08 (br m, 7H, C^2,6^-H *Ph*, C^2,5,6^-H *TA*), 7.31 (m, 1H, C^7^-H *TA*), 7.54 (d, *J* = 7.9 Hz, 1H, C^4^-H *TA*).

^13^C-NMR (500 MHz, MeOD): δ = 24.2, 24.4 (γC), 26.3 (***C***H_2_-Ar *TA*), 28.8, 29.0 [C(***C***H_3_)_3_], 30.1 (δC), 32.90 (βC), 33.3, 33.4 (β, δC), 39.9 (4×O-CH_2_-***C***H_2_-NH), 40.7 (εC *core*), 41.4 (CH_2_-NH *TA*), 42.2 (4×εC *L-Lys*), 55.5 (αC *core*), 56.1 (4×αC *L-Lys*), 67.8 (4×O-***C***H_2_-CH_2_-NH), 80.6 [***C***(CH_3_)_3_], 105.8, 106.0 (C^4^
*Ph*), 107.2, 107.5, 107.6 (C^2,6^
*Ph*), 112.3 (C^7^
*TA*), 113.1 (C^3^
*TA*), 119.3 (C^4^
*TA*), 119.6 (C^5^
*TA*), 122.3 (C^6^
*TA*), 123.6 (C^2^
*TA*), 128.7 (C^3a^
*TA*), 137.3, 137.9 (C^1^
*Ph*), 138.1 (C^7a^
*TA*), 157.8 [C=O (Boc)], 161.3 (C^3,5^
*Ph*), 169.4, 169.6 (CONH *Ph*), 174.3, 175.5 (CONH).

**[α]_D_^25^** = −22.7 (c 0.65, MeOH). 

**M.p.:** 128–131 °C.

#### 2.3.3. Dendrimer 18 (Obtained from 14)

Dendrimer 18 was obtained as white powder from 2.83 g (1.25 mmol) of dendrimer 14 dissolved in 50 mL MeOH after 7h hydrogenolysis, yield 97% (2.1 g).

C_88_H_144_O_19_N_16_, M = 1730.18 g/mol (monoisotopic mass 1729.1). LRMS (ESI, MeOH): 1752.4 [M + Na^+^]^+^, 1730.4 [M + H^+^]^+^, 865.8 [M + 2H^+^]^2+^ - *main signal*.

^1^H-NMR (500 MHz, MeOD): δ = 1.30–1.50 (br m, 12H, 5×γ, δCH_2_
*L-Lys and core*, δCH_2_
*core*) overlapped with 1.40, 1.42 [2s, 36H, 4×C(CH_3_)_3_], 1.58 [br m, 20H, 4×δCH_2_
*L-Lys*, 2×βCH_2_, 2×O-(CH_2_)_2_-C***H_2_***-C***H_2_***-CH_2_-NH], 1.75 [br m, 10H, 3×βCH_2_, 2×O-CH_2_-C***H_2_***-(CH_2_)_3_-NH], 2.75 (2m, 8H, 4×εCH_2_
*L-Lys*), 2.94 (t, *J* = 7.2 Hz, 2H, CH_2_-Ar *TA*), 3.21 [m, 4H, 2×O-(CH_2_)_4_-C***H_2_***-NH], 3.3 (m, 2H, εCH_2_
*core*), 3.41-3.67 (br m, 6H, 2×O-CH_2_-C***H_2_***-NH, C***H_2_***-NH *TA*), 3.89-4.09 (br m, 12H, 4×O-CH_2_, 4×αCH *L-Lys*), 4.48 (m, 1H, αCH *core*), 6.61 (m, 2H, C^4^-H *Ph*), 6.92-7.08 (br m, 7H, C^2,6^-H *Ph*, C^2,5,6^-H *TA*), 7.31 (m, 1H, C^7^-H *TA*), 7.54 (d, *J* = 7.85 Hz, 1H, C^4^-H *TA*).

^13^C-NMR (500 MHz, MeOD): δ = 24.1, 24.4 [γC, 2×O-(CH_2_)_2_-***C***H_2_-(CH_2_)_2_-NH], 26.3 (***C***H_2_-Ar *TA*), 28.8 [C(***C***H_3_)_3_], 29.9 [2×O-CH_2_-***C***H_2_-(CH_2_)_3_-NH], 30.1, 30.4 [δC, 2×O-(CH_2_)_3_-***C***H_2_-CH_2_-NH], 32.9, 33.1 (βC), 39.9 (2×O-CH_2_-***C***H_2_-NH), 40.2 [2×O-(CH_2_)_4_-***C***H_2_-NH], 40.6 (εC *core*), 41.3 (4×εC *L-Lys*), 41.4 (CH_2_-NH *TA*), 55.6 (αC *core*), 56.0, 56.1 (4×αC *L-Lys*), 67.8 (2×O-***C***H_2_-CH_2_-NH), 69.2 [2×O-***C***H_2_-(CH_2_)_4_-NH], 80.7 [***C***(CH_3_)_3_], 105.7, 106.1 (C^4^
*Ph*), 106.8, 107.2, 107.4 (C^2,6^
*Ph*), 112.3 (C^7^
*TA*), 113.1 (C^3^
*TA*), 119.3 (C^4^
*TA*), 119.6 (C^5^
*TA*), 122.3 (C^6^
*TA*), 123.6 (C^2^
*TA*), 128.7 (C^3a^
*TA*), 137.2, 137.8 (C^1^
*Ph*), 138.1 (C^7a^
*TA*), 157.8 [C=O (Boc)], 161.3, 161.7 (C^3,5^
*Ph*), 169.7, 169.8 (CONH *Ph*), 170.4, 174.3, 175.0, 175.3 (CONH).

**[α]_D_^25^** = −17.4 (c 1, MeOH). 

**M.p.:** 123–130 °C.

#### 2.3.4. Dendrimer 19 (Obtained from 15)

Dendrimer 19 was obtained from 1.02 g (0.46 mmol) of dendrimer 15 dissolved in 25 mL MeOH after 5h hydrogenolysis: yield 96.1% (0.74 g).

C_84_H_147_O_19_N_15_, M = 1671.16 g/mol (monoisotopic mass 1670.1). LRMS (ESI, MeOH): 1671.5 [M + H^+^]^+^, 836.0 [M + 2H^+^]^2+^ - *main signal*.

#### 2.3.5. Dendrimer 20 (Obtained from 16)

Dendrimer 20 was obtained in the coupling reaction of 4-(Boc-amino)benzoic acid [Boc-4-Abz-OH] (0.247 g, 1.04 mmol), HOBt (0.159 g, 1.04 mmol), EDC∙HCl (0.2 g, 1.04 mmol) with dendrimer 16 (0.4 g, 0.237 mmol), Et_3_N (0.26 mL, 1.9 mmol) in DMF (10 mL); obtained creamy powder, yield 60% (0.36 g). 

C_134_H_195_O_31_N_19_, M = 2568.09 g/mol (monoisotopic mass 2566.4). LRMS (ESI, MeOH): 2589.8 [M + Na^+^]^=^, 1306.3 [M + 2Na^+^]^2+^ - *main signal*, 1295.3 [M + H^+^ + Na^+^]^2+^, 1311.3 [M + MeOH + H^+^ + Na^+^]^2+^.

^1^H-NMR (500 MHz, MeOD): δ = 1.33–1.51 [br m, 16H, 5×γCH_2_
*L-Lys i core*, δCH_2_
*core*, 2×O-(CH_2_)_2_-C***H_2_***-(CH_2_)_2_-NH] overlapped with 1.38, 1.40 [2s, 36H, 4×C(CH_3_)_3_
*L-Lys*] i 1.50 [2s, 36H, 4×C(CH_3_)_3_
*PABA*], 1.61 [m, 16H, 4×δCH_2_
*L-Lys*, 2×β-CH_2_, 2×O-(CH_2_)_3_-C***H_2_***-CH_2_-NH], 1.72 [br m, 10H, 3×β-CH_2_, 2×O-CH_2_-C***H_2_***-(CH_2_)_3_-NH], 2.75 (t, *J* = 7.16 Hz, 2H, CH_2_-Ar *PEA*), 3.09-3.25 [2m, 6H, εCH_2_
*core*, 2×O-(CH_2_)_4_-C***H_2_***-NH], 3.34 (m, 9H, C***H_2_***-NH *PEA*, 4×εCH_2_
*L-Lys*), 3.48 (m, 3H, O-CH_2_-C***H_2_***-NH, C***H_2_***-NH *PEA*), 3.60 (m, 2H, O-CH_2_-C***H_2_***-NH), 3.91 [m, 4H, 2×O-C***H_2_***-(CH_2_)_4_-NH], 4.02 (m, 8H, 2×O-C***H_2_***-CH_2_-NH, 4×αCH *L-Lys*), 4.47 (m, 1H, αCH *core*), 6.60 (m, 2H, C^4^-H *Ph*), 6.92, 6.99 (2m, 4H, C^2,6^-H *Ph*), 7.10-7.23 (br m, 5H, C^2,3,4,5,6^-H *PEA*), 7.48 (m, 8H, C^3,5^-H *PABA*), 7.73 (m, 8H, C^2,6^-H *PABA*). 

^13^C-NMR (500 MHz, MeOD): δ = 24.3 (γC), 24.4 [2×O-(CH_2_)_2_-***C***H_2_-(CH_2_)_2_-NH], 24.5 (γC), 28.7 [C(***C***H_3_)_3_
*PABA*], 28.8 [C(***C***H_3_)_3_
*L-Lys*], 29.9 [2×O-CH_2_-***C***H_2_-(CH_2_)_3_-NH], 30.1, 30.2 [δC, 2×O-(CH_2_)_3_-***C***H_2_-CH_2_-NH], 32.8, 33.2 (βC), 36.5 (***C***H**_2_**-Ar *PEA*), 40.0 (2×O-CH_2_-***C***H_2_-NH), 40.2 [2×O-(CH_2_)_4_-***C***H_2_-NH] 40.5, 40.6 (εC), 42.0 (CH_2_-NH *PEA*), 55.6 (αC *core*), 56.1, 56.2 (4×αC *L-Lys*), 67.7 (2×O-***C***H_2_-CH_2_-NH), 69.2 [2×O-***C***H_2_-(CH_2_)_4_-NH], 80.6 [***C***(CH_3_)_3_
*L-Lys*], 81.2 [***C***(CH_3_)_3_
*PABA*], 105.8, 106.1 (C^4^
*Ph*), 106.7, 107.0, 107.2, 107.5 (C^2,6^
*Ph*), 118.8 (C^3,5^
*PABA*), 127.4 (C^4^
*PEA*), 129.2 (C^2,6^
*PABA*), 129.3 (C^1^
*PABA*), 129.5, 129.9 (C^2,3,5,6^
*PEA*), 137.2, 137.8 (C^1^
*Ph*), 140.4 (C^1^
*PEA*), 144.0 (C^4^
*PABA*), 154.8 [C=O (Boc) *PABA*], 157.8 [C=O (Boc) *L-Lys*], 161.2, 161.7 (C^3,5^
*Ph*), 169.6, 169.7 (CONH *Ph*), 169.8 (CONH *PABA*), 174.4, 175.1, 175.5 (CONH).

**[α]_D_^25^** = −12.7 (c 1, MeOH). 

**R_f_** = 0.38 (CHCl_3_/MeOH 8:1).

**M.p.:** 144–150 °C.

#### 2.3.6. Dendrimer 21 (Obtained from 17)

Dendrimer 21 was obtained in the coupling reaction of Boc-4-Abz-OH (0.218 g, 0.92 mmol), HOBt (0.141 g, 0.92 mmol), EDC∙HCl (0.176 g, 0.92 mmol), with dendrimer 17 (0.35 g, 0.21 mmol), Et_3_N (0.23 mL, 1.68 mmol) in DMF (10 mL); obtained creamy solid, yield 39.2% (0.21 g).

C_130_H_184_O_31_N_20_, M = 2522.97 g/mol (monoisotopic mass 2521.3). LRMS (ESI, MeOH): 2545.8 [M + Na^+^]^+^, 1284.0 [M + 2Na^+^]^2+^ - *main signal*.

^1^H-NMR (500 MHz, MeOD): δ = 1.34–1.43 (m, 10H, 5×γCH_2_
*L-Lys i core*) overlapped with 1.38 [s, 36H, 4×C(CH_3_)_3_
*L-Lys*], 1.53-1.66 (br m, 14H, 5×δCH_2_
*L-Lys i core*, 2×βCH_2_
*L-Lys*) overlapped with 1.50 [s, 36H, 4×C(CH_3_)_3_
*PABA*], 1.68-1.86 (br m, 6H, 3×βCH_2_
*L-Lys i core*), 2.91 (t, *J* = 7.1 Hz, 2H, CH_2_-Ar *TA*), 3.24 (m, 10H, 5×εCH_2_
*L-Lys i core*), 3.38-3.63 (br m, 10H, 4×O-CH_2_-C***H_2_***-NH, C***H_2_***-NH *TA*), 4.00 (br m, 12H, 4×O-C***H_2_***-CH_2_-NH, 4×αCH *L-Lys*), 4.48 (m, 1H, αCH *core*), 6.61 (m, 2H, C^4^-H *Ph*), 6.91-7.06 (br m, 7H, C^2,6^-H *Ph*, C^2,5,6^-H *TA*), 7.29 (d, *J* = 8.1 Hz, 1H, C^7^-H *TA*), 7.46 (dd, *J* = 8.8, 2.4 Hz, 8H, C^3,5^-H *PABA*), 7.51 (d, *J* = 7.9 Hz, 1H, C^4^-H *TA*), 7.72 (dd, *J* = 8.8, 2.4 Hz, 8H, C^2,6^-H *PABA*). 

^13^C-NMR (500 MHz, MeOD): δ = 24.2 (γC), 26.2 (***C***H_2_-Ar *TA*), 28.7 [C(***C***H_3_)_3_
*PABA*], 28.8 [C(***C***H_3_)_3_
*L-Lys*], 30.1, 30.2 (δC), 32.9, 33.2 (βC), 39.9 (4×O-CH_2_-***C***H_2_-NH), 40.5, 40.6 (εC), 41.4 (CH_2_-NH *TA*), 55.6 (αC *core*), 56.1 (4×αC *L-Lys*), 67.7 (4×O-***C***H_2_-CH_2_-NH), 80.6 [***C***(CH_3_)_3_
*L-Lys*], 81.2 [***C***(CH_3_)_3_
*PABA*], 105.8, 106.1 (C^4^
*Ph*), 107.2, 107.5 (C^2,6^
*Ph*), 112.3 (C^7^
*TA*), 113.4 (C^3^
*TA*), 118.8 (C^3,5^
*PABA*), 119.3 (C^4^
*TA*), 119.7 (C^5^
*TA*), 122.3 (C^6^
*TA*), 123.6 (C^2^
*TA*), 128.7 (C^3a^
*TA*), 129.2 (C^2,6^
*PABA*), 129.3 (C^1^
*PABA*), 137.3, 137.9 (C^1^
*Ph*), 138.1 (C^7a^
*TA*), 144.0 (C^4^
*PABA*), 154.8 [C=O (Boc) *PABA*], 157.8 [C=O (Boc) *L-Lys*], 161.2 (C^3,5^
*Ph*), 169.7 (CONH *Ph*), 174.3, 175.5 (CONH).

**[α]_D_^25^** = −14.7 (c 1, MeOH). 

**R_f_** = 0.38 (CHCl_3_/MeOH 8:1).

**M.p.:** 150–160 °C.

#### 2.3.7. Dendrimer 22 (Obtained from 18)

Dendrimer 22 was obtained in the coupling reaction of Boc-4-Abz-OH (0.24 g, 1.01 mmol), HOBt (0.155 g, 1.01 mmol), EDC∙HCl (0.194 g, 1.01 mmol), with dendrimer 18 (0.4 g, 0.23 mmol), Et_3_N (0.26 mL, 1.84 mmol) in DMF (10 mL), obtained yellow powder, yield 50% (0.3 g). 

C_136_H_196_O_31_N_20_, M = 2607.13 g/mol (monoisotopic mass 2605.4). LRMS (ESI, MeOH): 2628.5 [M + Na^+^]^+^, 1325.8 [M + 2Na^+^]^2+^ - *main signal*.

^1^H-NMR (500 MHz, MeOD): δ = 1.33–1.51 [br m, 16H, 5×γCH_2_
*L-Lys i core*, δCH_2_
*core*, 2×O-(CH_2_)_2_-C***H_2_***-(CH_2_)_2_-NH] overlapped with 1.38, 1.39 [2s, 36H, 4×C(CH_3_)_3_
*L-Lys*] i 1.50 [2s, 36H, 4×C(CH_3_)_3_
*PABA*], 1.54-1.87 [br m, 26H, 4×δCH_2_
*L-Lys*, 5×βCH_2_
*L-Lys i core*, 2×O-CH_2_-C***H_2_***-CH_2_-C***H_2_***-CH_2_-NH], 2.91 (t, *J* = 7.1 Hz, 2H, CH_2_-Ar *TA*), 3.09-3.24 [2m, 4H, 2×O-(CH_2_)_4_-C***H_2_***-NH], 3.31 (m, 10H, 5×εCH_2_
*L-Lys i core*), 3.39-3.64 (2br m, 6H, 2×O-CH_2_-C***H_2_***-NH, C***H_2_***-NH *TA*), 3.88 [m, 4H, 2×O-C***H_2_***-(CH_2_)_4_-NH], 4.00 (br m, 8H, 2×O-C***H_2_***-CH_2_-NH, 4×αCH *L-Lys*), 4.49 (m, 1H, αCH *core*), 6.57 (m, 2H, C^4^
*Ph*), 6.88-7.07 (br m, 7H, C^2,6^
*Ph*, C^2,5,6^-H *TA*), 7.29 (d, *J* = 8.1 Hz, 1H, C^7^-H *TA*), 7.48 (m, 8H, C^3,5^-H *PABA*), 7.51 (d, *J* = 8.0 Hz, 1H, C^4^-H *TA*), 7.72 (m, 8H, C^2,6^-H *PABA*).

^13^C-NMR (500 MHz, MeOD): δ = 24.2, 24.3 (γC), 24.4 [2×O-(CH_2_)_2_-***C***H_2_-(CH_2_)_2_-NH], 26.2 (***C***H_2_-Ar *TA*), 28.7 [C(***C***H_3_)_3_
*PABA*], 28.8, 29,0 [C(***C***H_3_)_3_
*L-Lys*], 29.9 [2×O-CH_2_-***C***H_2_-(CH_2_)_3_-NH], 30.1, 30.2 [2×O-(CH_2_)_3_-***C***H_2_-CH_2_-NH, δC], 32.9, 33.2 (βC), 40.0 (2×O-CH_2_-***C***H_2_-NH), 40.2 [2×O-(CH_2_)_4_-***C***H_2_-NH], 40.5, 40.7 (εC), 41.4 (CH_2_-NH *TA*), 55.6 (αC *core*), 56.1, 56.2 (4×αC *L-Lys*), 67.7 (2×O-***C***H_2_-CH_2_-NH), 69.2 [2×O-***C***H_2_-(CH_2_)_4_-NH], 80.6 [***C***(CH_3_)_3_
*L-Lys*], 81.2 [***C***(CH_3_)_3_
*PABA*], 105.8, 106.1 (C^4^
*Ph*), 106.7, 107.0, 107.2, 107.4 (C^2,6^
*Ph*), 112.3 (C^7^
*TA*), 113.1 (C^3^
*TA*), 118.8 (C^3,5^
*PABA*), 119.3 (C^4^
*TA*), 119.7 (C^5^
*TA*), 122.3 (C^6^
*TA*), 123.6 (C^2^
*TA*), 128.7 (C^3a^
*TA*), 129.2 (C^2,6^
*PABA*), 129.3 (C^1^
*PABA*), 137.8 (C^1^
*Ph*), 138.1 (C^7a^
*TA*), 144.0 (C^4^
*PABA*), 154.8 [C=O (Boc) *PABA*], 157.8 [C=O (Boc) *L-Lys*], 161.2, 161.7 (C^3,5^
*Ph*), 169.7 (CONH *Ph*), 169.8 (CONH *PABA*), 174.3, 175.1, 175.5 (CONH). 

**[α]_D_^25^** = −9.9 (c 1, MeOH). 

**R_f_** = 0.44 (CHCl_3_/MeOH 8:1).

**M.p.:** 149–155 °C. 

#### 2.3.8. Dendrimer 23 (Obtained from 19)

Dendrimer 23 was obtained in the coupling reaction of Boc-4-Abz-OH (0.427 g, 1.8 mmol), HOBt (0.276 g, 1.8 mmol), EDC∙HCl (0.345 g, 1.8 mmol), and dendrimer 19 (0.69 g, 0.41 mmol), Et_3_N (0.45 mL, 3.28 mmol) in DMF (15 mL); obtained white powder, yield 39.1% (0.41 g). LRMS (ESI, MeOH): 2569.6 [M + Na^+^]^+^, 1296.3 [M + 2Na^+^]^2+^ - *main signal*.

^1^H-NMR (500 MHz, DMSO): δ = 0.83 (br t, 3H, CH_3_
*DDA*), 1.18–1.50 (br m, 44H, 5×γ, δCH_2_
*L-Lys i core*, 2×βCH_2_, CH_2_-2-11 *DDA*) overlapped with 1.35 [s, 36H, 4×C(CH_3_)_3_
*L-Lys*] i 1.48 [s, 36H, 4×C(CH_3_)_3_
*PABA*], 1.55-1.78 (br m, 6H, 3×βCH_2_), 2.56-3.09 (m, 2H, εCH_2_), 3.19 (m, 10H, 4×εCH_2_, CH_2_-1 *DDA*), 3.30-3.48 (br m, 8H, 4×O-CH_2_-C***H_2_***-NH), 3.89 (m, 4H, 4×αCH *L-Lys*), 4.00 (m, 8H, 4×O-C***H_2_***-CH_2_-NH), 4.37 (m, 1H, αCH *core*), 6.62 (m, 2H, C^4^-H *Ph*), 6.77 (m, 4H, 4×αCH-N***H***
*L-Lys*), 7.00, 7.07 (2m, 4H, C^2,6^-H *Ph*), 7.50 (d, *J* = 8.7 Hz, 8H, C^3,5^-H *PABA*), 7.74 (d, *J* = 8.7 Hz, 8H, C^2,6^-H *PABA*), 7.86 (m, 1H, εCH_2_-N***H***
*core*), 7.98 (m, 4H, 4×O-CH_2_-CH_2_-N***H***), 8.24 (m, 4H, 4×εCH_2_-N***H***
*L-Lys*), 8.36 (2m, 2H, N***H***CH_2_-1 *DDA*, αCH-N***H***
*core*), 9.55 (br s, 4H, 4×C^4^-N***H***
*PABA*). 

^13^C-NMR (500 MHz, DMSO): δ = 13.9 (C^12^
*DDA*), 22.0, 23.0, 23.3 (γC_,_ C^11^
*DDA*), 26.2 (C^3^
*DDA*), 28.0, 28.1 [C(***C***H_3_)_3_], 28.6, 28.9-29.0 (δC, C^2^, C^4^-C^9^
*DDA*), 31.2, 31.5, 31.7 (βC, C^10^
*DDA*), 38.1 (4×O-CH_2_-***C***H_2_-NH), 38.4, 39.2 (εC, C^1^
*DDA*), 53.5 (αC *core*), 54.3 (4×αC *L-Lys*), 66.4, 66.5 (4×O-***C***H_2_-CH_2_-NH), 77.9 [***C***(CH_3_)_3_
*L-Lys*], 79.2, 79.3 [***C***(CH_3_)_3_
*PABA*], 103.7, 104.0 (C^4^
*Ph*), 105.9, 106.3 (C^2,6^
*Ph*), 117.0, 117.3 (C^3,5^
*PABA*), 127.8 (C^2,6^
*PABA*), 128.0 (C^1^
*PABA*), 136.2, 136.7 (C^1^
*Ph*), 142.0 (C^4^
*PABA*), 152.5, 152.6 [C=O (Boc) *PABA*], 155.3 [C=O (Boc) *L-Lys*], 159.3 (C^3,5^
*Ph*), 165.3, 165.6 (CONH *Ph*), 171.4, 172.5 (CONH). 

**[α]_D_^25^** = −12.0 (c 1, MeOH). 

**R_f_** = 0.77 (CHCl_3_/MeOH 8:1).

**M.p.:** 142–149 °C.

#### 2.3.9. *Boc*-Deprotected Dendrimers

General. A procedure for tert-butoxycarbonyl (*Boc*) group removal was performed using concentrated HCl and ethyl acetate using 0.15 - 0.32 g (0.059–0.126 mmol) of the respective dendrimer dissolved in 1 mL MeOH and 5–10 mL sat. HCl/AcOEt. Complete Boc group removal was detected after 1 h. Deprotected dendrimers 24–27 are in the form of octahydrochlorides.

#### 2.3.10. Dendrimer 24 (Obtained from 20)

0.26 g (0.101 mmol) of dendrimer 20 yielded pale yellow powder, yield 91.3% (0.19 g).

C_94_H_131_O_15_N_19_×8HCl, M = 2058.85 g/mol (monoisotopic mass 1766.0). LRMS (ESI, MeOH): 884.1 [M + 2H^+^]^2+^, 589.7 [M + 3H^+^]^3+^ - *main signal*.

^1^H NMR (500 MHz, MeOD): δ = 1.47 [br m, 16H, 5×γCH_2_
*L-Lys i core*, δCH_2_
*core*, 2×O-(CH_2_)_2_-C***H_2_***-(CH_2_)_2_-NH], 1.54–1.78 [br m, 16H, 4×δCH_2_
*L-Lys*, 2×O-CH_2_-C***H_2_***-CH_2_-C***H_2_***-CH_2_-NH], 1.80-1.97 (br m, 10H, 5×βCH_2_
*L-Lys i core*), 2.76 (t, *J* = 7.2 Hz, 2H, CH_2_-Ar *PEA*), 3.21 [br m, 4H, 2×O-(CH_2_)_4_-C***H_2_***-NH], 3.34 (m, 5H, C***H_2_***-NH *PEA*, 2×εCH_2_), 3.39 (t, *J* = 6.9 Hz, 6H, 3×εCH_2_), 3.46 (m, 1H, C***H_2_***-NH *PEA*), 3.54, 3.69 (2m, 4H, 2×O-CH_2_-C***H_2_***-NH), 3.85-4.00 [br m, 8H, 4×αCH *L-Lys*, 2×O-C***H_2_***-(CH_2_)_4_-NH], 4.09 (m, 4H, 2×O-C***H_2_***-CH_2_-NH), 4.45 (m, 1H, αCH *core*), 6.64 (m, 2H, C^4^-H *Ph*), 6.96, 7.02 (2m, 4H, C^2,6^-H *Ph*), 7.09-7.23 (br m, 5H, C^2,3,4,5,6^-H *PEA*), 7.50 (m, 8H, C^3,5^-H *PABA*), 7.99 (m, 8H, C^2,6^-H *PABA*).

^13^C NMR (500 MHz, MeOD): δ = 23.2, 23.4, 24.6 [γC, 2×O-(CH_2_)_2_-***C***H_2_-(CH_2_)_2_-NH], 29.9-30.0, 30.1 [δC, 2×O-CH_2_-***C***H_2_-CH_2_-***C***H_2_-CH_2_-NH], 32.3, 32.4, 32.8 (βC), 36.5 (***C***H**_2_**-Ar *PEA*), 40.2 (2×O-CH_2_-***C***H_2_-NH), 40.5 [εC, 2×O-(CH_2_)_4_-***C***H_2_-NH], 40.7 (εC), 42.0 (CH_2_-NH *PEA*), 54.5 (4×αC *L-Lys*), 55.8 (αC *core*), 67.6, 67.7 (2×O-***C***H_2_-CH_2_-NH), 69.3 [2×O-***C***H_2_-(CH_2_)_4_-NH], 105.8, 106.2 (C^4^
*Ph*), 106.8, 107.0, 107.3, 107.7 (C^2,6^
*Ph*), 124.4 (C^3,5^
*PABA*), 127.4 (C^4^
*PEA*), 129.5, 129.9 (C^2,3,5,6^
*PEA*), 130.4 (C^2,6^
*PABA*), 135.0 (C^1^
*PABA*), 136.3 (C^4^
*PABA*), 137.2, 137.8 (C^1^
*Ph*), 140.4 (C^1^
*PEA*), 161.2, 161.8 (C^3,5^
*Ph*), 168.4-168.5 (CONH *PABA*), 169.8 (CONH *Ph*), 170.0, 170.5, 174.5 (CONH).

**[α]_D_^25^** = −1.9 (c 1, MeOH).

**M.p.:** 175–191 °C. 

#### 2.3.11. Dendrimer 25 (Obtained from 21)

0.15 g (0.059 mmol) of dendrimer 21 yielded grey powder, yield 91.7% (0.11 g).

C_90_H_120_O_15_N_20_×8HCl, M = 2013.73 g/mol (monoisotopic mass 1720.9). LRMS (ESI, MeOH): 1721.8 [M + H^+^]^+^, 888.4 [M + MeOH + H^+^ + Na^+^]^2+^, 861.4 [M + 2H^+^]^2+^ - *main signal*, 592.6 [M + MeOH + 2H^+^ + Na^+^]^3+^, 574.6 [M + 3H^+^]^3+^.

^1^H-NMR (500 MHz, MeOD): δ = 1.45 (m, 10H, 5×γCH_2_
*L-Lys and core*), 1.62 (m, 10H, 5×δCH_2_
*L-Lys and core*), 1.71-1.96 (br m, 10H, 5×βCH_2_
*L-Lys and core*), 2.92 (m, 2H, CH_2_-Ar *TA*), 3.34 (m, 8H, 4×εCH_2_
*L-Lys*), 3.44 (m, 3H, εCH_2_
*core*, C***H_2_***-NH *TA*), 3.52 (m, 5H, 2×O-CH_2_-C***H_2_***-NH, C***H_2_***-NH *TA*), 3.68 (m, 4H, 2×O-CH_2_-C***H_2_***-NH), 3.92 (m, 4H, 4×αCH *L-Lys*), 4.08 (br m, 8H, 4×O-C***H_2_***-CH_2_-NH), 4.45 (m, 1H, αCH *core*), 6.67 (2m, 2H, C^4^-H *Ph*), 6.97-7.07 (br m, 7H, C^2,6^-H *Ph*, C^2,5,6^-H *TA*), 7.31 (d, *J* = 8.04 Hz, 1H, C^7^-H *TA*), 7.48 (m, 9H, C^4^-H *TA*, C^3,5^-H *PABA*), 7.98 (m, 8H, C^2,6^-H *PABA*).

^13^C-NMR (500 MHz, MeOD): δ = 23.2, 24.6 (γC), 26.2 (***C***H_2_-Ar *TA*), 30.0, 30.1 (δC), 32.3, 32.8 (βC), 40.2 (4×O-CH_2_-***C***H_2_-NH), 40.5, 40.7 (εC), 41.4 (CH_2_-NH *TA*), 54.5 (4×αC *L-Lys*), 55.9 (αC *core*), 67.6, 67.7 (4×O-***C***H_2_-CH_2_-NH), 105.9, 106.3 (C^4^
*Ph*), 107.3, 107.6 (C^2,6^
*Ph*), 112.3 (C^7^
*TA*), 113.1 (C^3^
*TA*), 119.3 (C^4^
*TA*), 119.6 (C^5^
*TA*), 122.3 (C^6^
*TA*), 123.6 (C^2^
*TA*), 124.3, 124.4 (C^3,5^
*PABA*), 128.7 (C^3a^
*TA*), 130.4 (C^2,6^
*PABA*), 135.0 (C^1^
*PABA*), 136.3 (C^4^
*PABA*), 137.4 (C^1^
*Ph*), 138.1 (C^7a^
*TA*), 161.2, 161.3 (C^3,5^
*Ph*), 168.5 (CONH *PABA*), 169.6 (CONH *Ph*), 170.5, 174.4 (CONH). 

**[α]_D_^25^** = −5.5 (c 1, MeOH). 

**M.p.:** 200–211 °C.

#### 2.3.12. Dendrimer 26 (Obtained from 22)

0.27 g (0.104 mmol) of dendrimer 22 yielded yellow powder, yield 96.8% (0.21 g).

C_96_H_132_O_15_N_20_×8HCl, M = 2097.89 g/mol (monoisotopic mass 1805.0). LRMS (ESI, MeOH): 930.4 [M + MeOH + H^+^ + Na^+^]^2+^, 925.4 [M + 2Na^+^]^2+^, 903.4 [M + 2H^+^]^2+^ - *main signal*, 620.6 [M + MeOH + 2H^+^ + Na^+^]^3+^, 602.6 [M + 3H^+^]^3+^. 

^1^H-NMR (500 MHz, MeOD): δ = 1.46 [br m, 14H, 5×γCH_2_
*L-Lys and core*, 2×O-(CH_2_)_3_-C***H_2_***-CH_2_-NH], 1.53–1.78 [br m, 18H, 5×δCH_2_
*L-Lys and core*, 2×O-CH_2_-(C***H_2_***)_2_-(CH_2_)_2_-NH], 1.88 (m, 10H, 5×βCH_2_
*L-Lys and core*), 2.92 (t, *J* = 7.1 Hz, 2H, CH_2_-Ar *TA*), 3.20 [m, 4H, 2×O-(CH_2_)_4_-C***H_2_***-NH], 3.34 (m, 4H, 2×εCH_2_), 3.39 (m, 7H, 3×εCH_2_, C***H_2_***-NH *TA*), 3.52 (m, 3H, O-CH_2_-C***H_2_***-NH, C***H_2_***-NH *TA*), 3.68 (m, 2H, 2×O-CH_2_-C***H_2_***-NH), 3.85-3.99 [br m, 8H, 4×αCH *L-Lys*, 2×O-C***H_2_***-(CH_2_)_4_-NH], 4.07 (m, 4H, 2×O-C***H_2_***-CH_2_-NH), 4.46 (m, 1H, αCH *core*), 6.60, 6.65 (2m, 2H, C^4^-H *Ph*), 6.91-7.07 (br m, 7H, C^2,6^-H *Ph*, C^2,5,6^-H *TA*), 7.31 (d, *J* = 8.1 Hz, 1H, C^7^-H *TA*), 7.50 (m, 9H, C^4^-H *TA*, C^3,5^-H *PABA*), 7.98 (m, 8H, C^2,6^-H *PABA*).

^13^C-NMR (500 MHz, MeOD): δ = 23.2, 23.3, 24.5 [γC, 2×O-(CH_2_)_2_-***C***H_2_-(CH_2_)_2_-NH], 26.2 (***C***H_2_-Ar *TA*), 29.9, 30.0, 30.1 [δC, 2×O-CH_2_-***C***H_2_-CH_2_-***C***H_2_-CH_2_-NH], 32.3, 32.4, 32.9 (βC), 40.2 (2×O-CH_2_-***C***H_2_-NH), 40.5 [εC, 2×O-(CH_2_)_4_-***C***H_2_-NH], 40.7 (εC), 41.4 (CH_2_-NH *TA*), 54.4, 54.5 (4×αC *L-Lys*), 55.8 (αC *core*), 67.6 (2×O-***C***H_2_-CH_2_-NH), 69.2, 69.3 [2×O-***C***H_2_-(CH_2_)_4_-NH], 105.8, 106.2 (C^4^
*Ph*), 106.8, 107.0, 107.3, 107.6 (C^2,6^
*Ph*), 112.3 (C^7^
*TA*), 113.1 (C^3^
*TA*), 119.3 (C^4^
*TA*), 119.6 (C^5^
*TA*), 122.3 (C^6^
*TA*), 123.6 (C^2^
*TA*), 124.3 (C^3,5^
*PABA*), 128.7 (C^3a^
*TA*), 130.3 (C^2,6^
*PABA*), 135.0 (C^1^
*PABA*), 136.3 (C^4^
*PABA*), 137.2, 137.8 (C^1^
*Ph*), 138.1 (C^7a^
*TA*), 161.2, 161.8 (C^3,5^
*Ph*), 168.5 (CONH *PABA*), 169.8 (CONH *Ph*), 170.0, 170.5, 174.4 (CONH). 

**[α]_D_^25^** = +0.06 (c 1, MeOH). 

**M.p.:** 189–196 °C. 

#### 2.3.13. Dendrimer 27 (Obtained from 24)

0.32 g (0.126 mmol) of dendrimer 24 yielded creamy powder, yield 97.7% (0.25 g).

C_92_H_135_O_15_N_19_×8HCl, M = 2038.86 g/mol (monoisotopic mass 1746.0). LRMS (ESI, MeOH): 874.1 [M + 2H^+^]^2+^ - *main signal*, 583.0 [M + 3H^+^]^3+^. 

^1^H NMR (500 MHz, DMSO): δ = 0.87 (br t, 3H, CH_3_
*DDA*), 1.22–1.33 (br m, 18H, CH_2_-3-11 *DDA*), 1.47 (m, 12H, 5×γCH_2_
*L-Lys and core*, CH_2_-2 *DDA*), 1.63 (m, 10H, 5×δCH_2_
*L-Lys and core*), 1.89 (m, 10H, 5×βCH_2_
*L-Lys and core*), 3.16 (m, 2H, CH_2_-1 *DDA*), 3.33 (m, 8H, 4×εCH_2_
*L-Lys*), 3.46 (m, 2H, εCH_2_
*core*), 3.54, 3.68 (2m, 8H, 4×O-CH_2_-C***H_2_***-NH), 3.93 (m, 4H, 4×αCH *L-Lys*), 4.10 (m, 8H, 4×O-C***H_2_***-CH_2_-NH), 4.47 (m, 1H, αCH *core*), 6.68 (m, 2H, C^4^-H *Ph*), 6.99, 7.05 (2m, 4H, C^2,6^-H *Ph*), 7.50 (dd, *J* = 8.5, 2.65 Hz, 8H, C^3,5^-H *PABA*), 7.99 (m, 8H, C^2,6^-H *PABA*).

^13^C NMR (500 MHz, DMSO): δ = 14.4 (C^12^
*DDA*), 23.2, 23.7, 24.7 (γC_,_ C^11^
*DDA*), 28.0 (C^3^
*DDA*), 29.9, 30.1 (δC, C^2^
*DDA*), 30.4-30.8 (C^4^-C^9^
*DDA*), 32.3, 32.9 (βC), 33.0 (C^10^
*DDA*), 40.2 (4×O-CH_2_-***C***H_2_-NH), 40.5 (εC *core*, C^1^
*DDA*), 40.8 (4×εC *L-Lys*), 54.5 (4×αC *L-Lys*), 55.9 (αC *core*), 67.7 (4×O-***C***H_2_-CH_2_-NH), 106.0, 106.3 (C^4^
*Ph*), 107.3, 107.6 (C^2,6^
*Ph*), 124.4, 124.7 (C^3,5^
*PABA*), 129.9, 130.4 (C^2,6^
*PABA*), 134.9 (C^1^
*PABA*), 136.3 (C^4^
*PABA*), 137.3, 137.9 (C^1^
*Ph*), 161.2, 161.3 (C^3,5^
*Ph*), 168.5 (CONH *PABA*), 169.6 (CONH *Ph*), 170.5, 174.4 (CONH). 

**[α]_D_^25^** = −4.7 (c 1, MeOH). 

**M.p.:** 182–199 °C.

### 2.4. Primary Cultures of Cerebellar Granule Cells (CGC)

Primary CGC cultures were prepared from 7-day-old Wistar rats of both sexes of the outbred stock CmD: (WI)WU. A standard previously described procedure for isolating and culturing cerebellar granule cells was used [23,24,25]. Procedures using rat pups were performed in accordance with international standards of animal care guidelines and were approved by the Local Care of Experimental Animals Committee. Cerebellar slices were trypsinized and triturated in basal Eagle’s medium (GIBCO, Deisenhofen, Germany) supplemented with 10% fetal calf serum, 25 mM KCl, 4 mM glutamine, penicillin (50 U/mL) and streptomycin (50mg/mL) to obtain cell suspensions. Cells were then seeded on plates coated with poly-L-lysine. Density of the cell suspension seeded on the 24-well plates (NUNC) was 1 × 10^6^ cells per well for measurements of 2′,7-dichlorofluorescein (DCF) fluorescence and to evaluate CGC viability. Cultures were treated with 7.5 µM cytosine arabinofuranoside 36 h after plating to prevent the replication of non-neuronal cells. After 7 days of culture in vitro, the cultures were used for experiments.

#### 2.4.1. Cerebellar Granule Cells Viability

The effect of the tested dendrimers alone or together with glutamate (Glu) on CGC viability was assessed 24 h after a 30 min exposure of the cultures to the test substances using propidium iodide staining. The culture medium was replaced with Locke 25 buffer containing 134 mM NaCl, 25 mM KCl, 2.3 mM CaCl_2_, 4 mM NaHCO_3_, 5 mM HEPES (pH 7.4), 5 mM glucose, and freshly prepared solutions of the test substances in concentration: 0.2, 2 and 20 µM, 100 µM Glu or vehicles (0.2% DMSO), as required. After a 30 min incubation at 37 °C and washes with Locke 25 buffer, the original growth medium was replaced and CGCs were cultured for an additional 24 h. Then, the cells were fixed with 80% methanol, stained with propidium iodide (PI) (0.5 µg/mL), and viable and dead neurons were counted using an Axiovert fluorescence microscope (Carl Zeiss AG, Oberkochen, Germany), with a user who was blinded to the experimental groups. The viability of the neurons was determined as percentages of live cells in proportion to all cells.

#### 2.4.2. Reactive Oxygen Species (ROS) Measurement

Reactive oxygen species (ROS) production in CGCs was monitored by measuring the fluorescence of DCF, a product of the ROS-mediated cleavage and oxidation of the precursor molecule DCFH-DA ((2′,7′-dichlorodihydrofluorescein diacetate, Invitrogen Molecular Probes, Waltham, MA USA) that easily penetrates cells. The experiments were preceded by loading the cells with a fluorescent probe. For this purpose, CGC cultures were incubated with the original culture medium containing 100 mM DCFH-DA for 30 min at 37 °C. Then, the cultures were washed 3 times with Locke 5 buffer containing 154 mM NaCl, 5 mM KCl, 2.3 mM CaCl_2_, 4 mM NaHCO_3_, 5 mM glucose and 5 mM HEPES (pH 7.4). Next, the cells were incubated with Locke 5 buffer and the fluorescence of the cell-entrapped probes was measured using a microplate reader FLUOstar Omega (Ortenberg, Germany) at 485 nm excitation and 538 nm emission wavelengths. After determining the baseline fluorescence of the cells incubated in Locke 5 buffer, changes in the fluorescence intensity were recorded every 60 s. As a positive control was used 10 µM H_2_O_2_ was added to the CGC cultures after 5^th^ min as well as tested dendrimers. The results are presented as percent changes in fluorescence intensity relative to the basal level (F/Fo %) vs. the duration of the measurement after the addition of the test compounds on a microplate reader [24]. 

### 2.5. Cell Proliferation Assays

The established human melanoma cell line MeW155 and human fibroblast AW/4, were obtained from the institutional repository at the Maria Skłodowska-Curie Memorial Institute and Oncology Centre in Warsaw. The melanoma and the fibroblast cell lines were cultured in Eagle’s 1959 minimal essential medium (MEM) (Biomed-Lublin, Lublin, Poland) supplemented with 10% fetal calf serum (Sigma Aldrich Chemical Co., USA), 50 mg/mL penicillin G, 50 mg/mL streptomycin, and 0.1 % glutamine in a humidified atmosphere with 5% CO_2_. Adherent cell cultures were plated in 24-well plates. The initial cell number in culture and timing of medium supplementation was adjusted for each cell line to achieve the optimal cell growth. The culture timing was adjusted to harvest cells at the state of sub-confluency. Adherent cells were harvested using 0.25% trypsin with EDTA (Invitrogen, Waltham, MA USA) in concentration 200 mg/L. Following trypsinization, the harvested cells were counted directly in a hemacytometer under a microscope, and cell viability was evaluated according to trypan blue staining. 

#### 2.5.1. DPPH Assay

Chemicals: Trolox^®^ (6-hydroxy-2,5,7,8-tetramethylchroman-2-carboxylic acid; Sigma-Aldrich, Saint Louis, MO, USA) was used as an antioxidant standard. Trolox (3.3 mM) was prepared in methanol for use as a stock standard. DPPH^●^, 2,2′-diphenyl-1-picrylhydrazyl radical (Sigma Aldrich, Steinheim, Germany) was dissolved in methanol. Assay: Experiments were performed according to Barton et al., with small modifications [26]. The obtained solution of DPPH^●^ in methanol had a concentration of 1.6 mM and was incubated for 2 h before measurement at room temperature in the dark. For the studies of antioxidant properties, the DPPH radical solution was diluted with methanol to an absorbance of 1.00 at 517 nm.

Studied compounds (final concentrations 0.001–0.014 μM) or Trolox (final concentration 0.003–0.039 mM) were added to diluted DPPH radical solution. After 15 min, incubation absorbance was read using the spectrophotometer. Results of the ability of dendrimers to scavenge of radical DPPH are presented as Trolox equivalent antioxidant capacity (TEAC). In the concentration range investigated, 50 % of radical scavenge (IC_50_) was not achieved. All experiments were repeated twice.

#### 2.5.2. ABTS Assay 

Chemicals: Trolox® (6-hydroxy-2,5,7,8-tetramethylchroman-2-carboxylic acid) was used as an antioxidant standard. Trolox (4.2 mM) was prepared in methanol for use as a stock standard. ABTS, 2,2′-azinobis(3-ethylbenzothiazoline-6-sulfonic acid, Sigma-Aldrich, Saint Louis, MO, USA) di-ammonium salt was dissolved in potassium persulfate (Sigma Aldrich, Steinheim, Germany).

Assay: experiments were performed according to Pellegrini et al., [27] Potassium persulfate was dissolved in water to a final concentration of 2.45 mM. ABTS was dissolved in the obtained earlier solution of potassium persulfate. The obtained solution 1.8 mM ABTS^•+^ radical cation was incubated for 16 hours at room temperature in the dark before use. For the study of dendrimers, the ABTS radical solution was diluted with water to an absorbance of 1.00 at 734 nm.

Studied dendrimers (final concentrations 0.001–0.014 μM) or Trolox (final concentration 0–0.03 mM) were added to diluted ABTS^•+^ solution and the absorbance reading was taken 10 min after mixing using an automated ultraviolet–visible (UV–Vis) Carry 100E spectrophotometer (Varian, Mulgrave, Victoria, Australia). In this concentration range IC_50_ (μM) was defined only for Trolox (IC_50_ value is the concentration of the sample required to inhibit 50% of radical). Results are presented as the ability of dendrimers to scavenge free cationic radical ABTS^•+^ in relation to Trolox and expressed as TEAC (Trolox equivalent antioxidant capacity). Parameters IC_50_ (μM) and TEAC (μM) were determined with a relative uncertainty of less than five percent. All analyses were performed in triplicate and the results were expressed as the mean value ± standard deviation.

## 3. Results

### 3.1. Antioxidant Properties—Neutral (DPPH) and Cationic Radicals (ABTS) Quenching Potency

The dendrimers were investigated for concentration-dependent in vitro free neutral radical scavenging ability assay that shows capability of a compound to interact with stable neutral radicals prepared from DPPH and with free cationic radicals prepared from ABTS. Both applied tests are presenting different aspects of antioxidative mechanism. While the DPPH assay shows the capacity of antioxidant to transfer electrons, the ABTS assay determines cationic free radical scavenging activity involving both electron and hydrogen transfer mechanisms. In both tests, the reference compound is Trolox, known radicals scavenger. The results of DPPH and ABTS tests for dendrimers at concentrations 20, 100 and 180 μM are shown in Figure 1. Dendrimers show different scavenging capacity in both tests, being better scavengers of radical cations than neutral radicals. The highest cationic radical scavenging ability in ABTS test was found for dendrimers D25 (8.9) and D26, bearing indole moiety, whereas the highest neutral radical scavenging in DPPH test was shown by and D24 (5.78) and D27 with the respective phenylethylamine or dodecylamine moiety located at C-terminus. Although scavenging level was enhanced upon increasing dendrimer concentrations and shows significant dose-effect, no simple additivity was detected. 

### 3.2. Cytotoxicity on Melanoma and Fibroblast Cell Lines 

Skin, the organ most exposed to the external stimuli coming from everyday environment (UV radiation, air, earth, and water pollution, etc.) is capable of eliminating the undesired effects of oxidation damage and reducing health risk up to a certain level [28]. Therefore, the interactions of skin cells with natural or synthetic compounds bearing groups known for antioxidant effect is at present of continuous interest. To evaluate the effects of dendrimers 24–27 on the proliferation of skin malignant cells, the human melanoma MeW155 cells were treated for 24 h with different concentrations of dendrimers (0, 10, 20, and 100 µM). The proliferation rate of the treated groups at 100 µM was significantly reduced while being still observed at 20 µM (except for dendrimer 25), when compared with that of the control group (0 µg/mL of the respective dendrimer) as demonstrated in Figure 2.

On the other hand, dendrimers 24–27 at 20 μM concentration that causes melanoma cells’ growth inhibition (except of D25) expressed very little toxicity or even growth promoting activity in human AW/4 fibroblast cells, as demonstrated in Figure 3. Remarkably, dendrimer D24 at this concentration shows cytotoxicity against MeW155 and promotes growth of fibroblast AW/4 cells. 

### 3.3. Toxicity of Dendrimers on Primary Cultures of Cerebellar Granule Cells 

The primary rat cerebellar granule cells (CGC) were cultivated and cell viability in the presence of dendrimers was measured for ethidium bromide (EB)-stained samples. Figure 4A presents viability of CGC after 30 min incubation with dendrimers D24–D27 in three different concentrations. Tested substances were diluted in DMSO, that was kept at 0.2% level in all used dendrimer concentrations (0.2, 2 and 20 µM). The live and dead neurons, which differ in morphology, were assessed 24 h later using PI staining and fluorescent microscopy. Vehicle (0.2% DMSO) did not change the viability of the cells. Dendrimers D24 and D25 were not toxic in all used concentrations; whereas dendrimer D27 was toxic only in the highest dose (20 µM), when viability decreased to 9%. Dendrimer D26 is the most toxic of the all tested substances. The CGC viability decreased to: 48%, 24 % and 6 % for 0.2, 2 and 20 μM concentration, respectively.

To assess the potential impact of dendrimers in the presence of the main neurotransmitter—glutamate (Glu)—in the context of its excitotoxicity on neurons, D24–D27 were incubated with 100 µM Glu (control) and a mixture of 100 µM Glu with the dendrimers in the two lowest concentrations, 0.2 and 2.0 μM (Figure 4B). Dendrimer D26 was excluded from this experiment, because of its toxicity. The 30 min. incubation with 100 µM Glu decreased CGC viability in the control from 94% to 52%. Addition of D25 at both concentrations to the medium with Glu had no effect on CGC viability, as compared to Glu alone. However, incubation with D24 in 2 µM concentration just before Glu addition resulted in an increase of the CGC viability by 17% (from 52% to 61%). Even more visible is effect for D27, where dendrimer in concentration of 0.2 or 2 µM evoked an increase in the number of living cells from 52% to 63% and 66%, respectively. 

An explanation of this phenomenon might be proposed at the supramolecular level. Evidently the tested cationic dendrimers may form salts with anionic glutamate dissolved in Locke medium, which in turn decrease the effective concentration of Glu, diminishing its excitotoxicity on neurons. However, the formation of salts cannot be the only explanation of the small but statistically relevant increase in CGC cells proliferation since an excess of Glu vs. dendrimers’ concentration is still very high (500- or 50-fold). For example, a ca. 10 % increase in cell viability is observed even if D27 is present at the lowest concentration 0.2 µM.

### 3.4. Effect of Dendrimers on the Reactive Oxygen Species Production in Cerebral Granule Cells Cultures

To acquire an information on the potential influence of PABA-derivatized dendrimers on ROS production in CGCs, the number of free radicals was measured using fluorescent probe DCF-DA (Figure 5A–E). The effect was tested for three different concentrations of dendrimers: 0.2, 2 and 20 µM. As evidenced by the increase of DCF-DA fluorescence in comparison to the control, all tested compounds enhanced dose-dependent ROS production in CGC neurons. Dendrimer D26 was the most harmful, and even at the lowest concentration (0.2 µM) evoked significant DCF fluorescence from 101% in DMSO to 119% in the 35th min of experiment. Increasing amounts of D26 to 2 and 20 µM potentiated DCF fluorescence by 23% and 77%, respectively. 

However, as shown in Figure 5E, the effect of tested dendrimers on ROS production in CGC is much smaller in comparison to that evoked by 10 µM H_2_O_2_ when DCF fluorescence reached 473% of the control.

Dendrimers D24 and D27 were the most neutral in this test. The DCF fluorescence for their highest concentration (20 µM) was increased up to 120 % and 127 % of the control level, respectively. Dendrimers D25 and D26 that are the most potent antioxidants produced also up to ca. 75% more of the reactive oxygen species. Dendrimer D25 increased DCF fluorescence up to 124% of the control value when was in 2 µM concentration, while used in 20 µM increased ROS production by 76%. 

## 4. Discussion

Each of the studied dendrimers contains in their structure several chemical groups that might evoke toxicity in the living cells with less impact on the in vitro laboratory testing. First, these molecules are multiple cations due to the (+8) charge located on four aliphatic and four aromatic amino groups converted to hydrochloride form (i.e. they are formally protonated). Cationic molecules may affect cell membrane integrity due to electrostatic interactions with negatively charged phospholipid head groups. On the other hand, they can easily permeate cells and might be used as a carrier system. Moreover, each dendrimer contains four redox-active PABA moieties and one chemically different chemical residue (phenylethylamine, tryptamine or dodecylamine), located at C-terminal positions, that may also influence the response of biological targets. Among those, the aliphatic dodecyl chain that is often facilitating hydrophobic interactions with the phospholipid tails, is probably the most harmful for the cell’s membrane. Dendrimers D25 and D26 contain redox active indole moiety and are constructed from the same residues. The only difference is in combination of arm lengths—D25 has four short (*n* = 1) arms, while D26 has two short and two long arms (*n* = 4).

The rationally designed dendrimers with multiple redox-active PABA residues were tested for antioxidative properties on two levels - molecular and cellular. 

The weak antioxidative power of PABA found in ABTS and DPPH laboratory tests is probably due to the fact that this is, to the large extent, a photoactive biomolecule which generates radicals mostly by UV irradiation. In contrast, dendrimers carrying multiple copies of PABA show a significant radical scavenging capacity with better predisposition for quenching radical cations. High antioxidant activity of dendrimers D25 and D26 in both tests probably shows an additional contribution from indole residue that is known to form cationic and neutral radicals [29,30]. 

On the cellular level, despite the slight generation of ROS (up to 125% of control) in concentrations below 20 μM, dendrimers did not markedly influence the viability of CGC cells. The only exception is D26, which reduced the number of healthy cells in a dose-dependent manner; by 50% at the lowest tested concentration (0.2 µM) and by 94% at concentration of 20 µM. Such high toxicity (but still about 6 times smaller than that produced by 10 µM H_2_O_2_) is unexpected, since D26 and D25 have the same essential residues and generated ROS and quenched radicals on the same level. The fact that their structure differs only in dendrimer arm lengths strongly suggests that the ROS generation is not directly connected with neuronal cells toxicity under our in vitro conditions and it is the dendrimer structure that “presents” to the cell membrane or cell organelles the potentially toxic groups in various way, (e.g., cationic amino groups). Among other tested dendrimers, only D27 (carrying at the C-terminus dodecyl chain) at highest concentration of 20 µM significantly reduced CGC and melanoma cell viability what might be correlated with statistically significant ROS generation. On the other hand, as it was shown earlier by Zieminska et al. [31] that 45 min incubation of CGC with 10 or 25 µM H_2_O_2_ did not influence cells viability significantly. 

The observed increase of the number of fibroblast cells upon addition of 20 μM of D24 and to the lesser extend D25 and positive impact of D24 and D27 on the viability of CGC cells under Glu-induced stress suggests that in certain chemical arrangements dendrimers containing PABA residue, which is an essential fragment in the structure of the folic acid molecule, are taken up more readily by malignant or stressed cells. 

In summary, the designed PABA-functionalized peptide dendrimers might be a potential source of new antioxidants with cationic and neutral radicals scavenging potency and/or new compounds with marked selectivity against human melanoma cells or glutamate-stressed CGC neurons. Significant enhancement of antioxidant potency of these dendrimers in comparison with the potency of the single PABA moiety might be related to their specific 3D structure and perhaps accounts for the so called “dendrimeric effect”. Although dendrimers produce in neuron cells a small amount of ROS, they provide a relatively safe carrier system that, probably due to their polycationic structure, can permeate membranes of neural and other cells. Interestingly, all dendrimers at 0.2–20 µM concentration (except one) increased the percentage of viable fibroblasts and CGC cells treated with 100 μM glutamate. However, further studies of these redox active compounds against other cancer cell lines are necessary, to confirm if, by analogy with natural antioxidants (e.g., polyphenols like chlorogenic or caffeic acids, curcumin, etc.), they are not only radical scavengers but also have the ability to destroy other types of cancer cells in the fashion of ROS-inducing drugs. It should be also emphasized that the present study provides once again evidence that the specific structure of the dendrimer tree can have a significant influence on drug–target interactions.

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
