# Peer review of "Molecular Antioxidant Properties and In Vitro Cell Toxicity of the p-Aminobenzoic Acid (PABA) Functionalized Peptide Dendrimers §"

_biomolecules, 2019, doi:10.3390/biom9030089_

Reviewer 1 Report

The manuscript explains the design of a group of peptide dendrimers carrying multiple copies of PABA and their evaluation for antioxidant properties, cytotoxicity against human melanoma and fibroblast cells as well as against primary cerebral granule cells, etc. The manuscript is written well and contains novel data and results. I recommend publication of this article.

Author Response

Thank you.

Reviewer 2 Report

The manuscript entitled " Molecular antioxidant properties and in vitro cell

toxicity of the p-aminobenzoic acid (PABA) functionalized peptide dendrimers" by Dr. Zofia Urbanczyk-Lipkowska et al, recommended for publication after the following comments are addressed.

v  In scheme 2 legend not described the reaction in the scheme and its very confusing. I suggest modifying the text and reaction conditions in the legend.

v  In line 144, (30-23) this is not correct.

v  In line 156 and 157 structure of the molecule with chemical shifts is not suitable for manuscript. I suggest removing that from manuscript.

v  In scheme 3, roman number 3 refer to reaction conditions should be correct as roman number 2.

v  In scheme 3 after deprotection there no HCl salt with product due to neutral conditions are using for deprotection.

v  In Figure 2 and 3 , I suggest to remove the grid lines.

v  In Figure 2 and 3 , I suggest to put the data in ug/mL, which reflects the real activity of dendrimers as they are big molecules.

v  I suggest to rewrite the conclusion, because the conclusion in the manuscript not summarized and described results obtained in the project.

Author Response

The manuscript entitled " Molecular antioxidant properties and in vitro cell

toxicity of the p-aminobenzoic acid (PABA) functionalized peptide dendrimers" by Dr. Zofia Urbanczyk-Lipkowska et al, recommended for publication after the following comments are addressed.

 v  In scheme 2 legend not described the reaction in the scheme and its very confusing. I suggest modifying the text and reaction conditions in the legend.

 Response: As suggested, reaction conditions for reactions described previously in ref. 21 were removed.

v  In line 144, (30-23) this is not correct.

Response: Numbering of compounds were corrected to 2027

v  In line 156 and 157 structure of the molecule with chemical shifts is not suitable for manuscript. I suggest removing that from manuscript.

 As suggested, Figure 1 was removed from the manuscript. Figures numbering was changed, accordingly.

v  In scheme 3, roman number 3 refer to reaction conditions should be correct as roman number 2.

 Corrected

In scheme 3 after deprotection there no HCl salt with product due to neutral conditions are using for deprotection.

 Deprotection reaction performed in these conditions leads to HCl salts of the final dendrimers. Such form might be beneficial for their biological properties. It is well documented that cationic compounds permeate cellular membranes easier.

v  In Figure 2 and 3 , I suggest to remove the grid lines.

 Grid lines were removed.

v  In Figure 2 and 3 , I suggest to put the data in ug/mL, which reflects the real activity of dendrimers as they are big molecules.

We would like to retain micromoles in Figure 2 and 3 in order to keep the same units as in the whole manuscript and ease discussion.

v  I suggest to rewrite the conclusion, because the conclusion in the manuscript not summarized and described results obtained in the project.

Discussion and conclusions part has been rewritten to achieve consistency with the data presented in experimental part.

Reviewer 3 Report

New modified peptides with biologic activity and low toxicity are desired. This manuscript is describing the obtaining of a new class of  p-aminobenzoic acid based peptides dendrimers with antioxidant properties  and low toxicity of the tested living cells.

Firstly, the authors reported on the design, synthesis and characterization (MS, 1H NMR, 13C NMR, m.p.) of dendrimers and PABA-peptide dendrimers. Thereafter, the viability of the dendrimers D24-D27 was evaluated in cerebellar granule cells of rats, and in human cell lines: melanoma MeW155 and fibroblast AW/4. Then the antioxidant activity of the dendrimers was evaluated by using DPPH and ABTS assays.

In general the data are strong, and convincingly shows the efficiency of the new class of PABA-based peptide dendrimers as antioxidant molecules. The effect of this compounds was tested in diverse living cells (rat cells: CGC, human cells: melanoma and fibroblast), and the controls are suitable used. The viability of the cells depended in a dose manner, D26 being the most toxic in CGC cells. The manuscript is well written, concise and the appropriate analyses are performed.

Overall, this is a well performed study that I consider that is important and represent a new strategy to conveniently obtain compounds with antioxidant properties, as it was demonstrated by DPPH and ABTS assays.

1.Please rewrite the following phrases:

- lines 57-58: “One of the emerging medical problems is the impact an increased exposition to ozone level and ultra-violet radiation on public health.”

- lines 160-161: “(b) in dendrimers 24 27, measured in MeOD (for dendrimers for 24 and 27 measured in DMSO some δ values are slightly different). “

-lines 602-603: “Viability in control and cells treated with 0.2 % DMSO was the same - 94 %.”

2. Please take into consideration to be constant in showing the results related to toxicity of the dendrimers on cells: choose table or graph and reconsider Table 2 or Figure 3. Also please include statistical data.

3. The manuscript may be checked by an English native speaker.

Author Response

New modified peptides with biologic activity and low toxicity are desired. This manuscript is describing the obtaining of a new class of  p-aminobenzoic acid based peptides dendrimers with antioxidant properties  and low toxicity of the tested living cells.

Firstly, the authors reported on the design, synthesis and characterization (MS, 1H NMR, 13C NMR, m.p.) of dendrimers and PABA-peptide dendrimers. Thereafter, the viability of the dendrimers D24-D27 was evaluated in cerebellar granule cells of rats, and in human cell lines: melanoma MeW155 and fibroblast AW/4. Then the antioxidant activity of the dendrimers was evaluated by using DPPH and ABTS assays.

In general the data are strong, and convincingly shows the efficiency of the new class of PABA-based peptide dendrimers as antioxidant molecules. The effect of this compounds was tested in diverse living cells (rat cells: CGC, human cells: melanoma and fibroblast), and the controls are suitable used. The viability of the cells depended in a dose manner, D26 being the most toxic in CGC cells. The manuscript is well written, concise and the appropriate analyses are performed.

Overall, this is a well performed study that I consider that is important and represent a new strategy to conveniently obtain compounds with antioxidant properties, as it was demonstrated by DPPH and ABTS assays.

1.Please rewrite the following phrases:

- lines 57-58: “One of the emerging medical problems is the impact an increased exposition to ozone level and ultra-violet radiation on public health."

Changed to: “One of the emerging medical problems is diminishing negative effects of an increased exposition of human population to ozone level and ultra-violet radiation.”

- lines 160-161: “(b) in dendrimers 24 27, measured in MeOD (for dendrimers for 24 and 27 measured in DMSO some δ values are slightly different). “

As suggested also by another referee, Figure 1 and therefore, also Figure caption were removed from the text.

-lines 602-603: “Viability in control and cells treated with 0.2 % DMSO was the same - 94 %.”

 This information was replaced by “Vehicle (0.2 % DMSO) did not change the viability of the cells.”

2. Please take into consideration to be constant in showing the results related to toxicity of the dendrimers on cells: choose table or graph and reconsider Table 2 or Figure 3. Also please include statistical data.

Following referee’s suggestion, all the mentioned data were presented in a graph mode. Appropriate correction in the text was done and  statistical data are added.

3. The manuscript may be checked by an English native speaker.

Response: After introducing suggested corrections two last chapters were sent again for editing by the native speaker.